# Punching Shear Behavior of Slabs Made from Different Types of Concrete Internally Reinforced with SHCC-Filled Steel Tubes

**DOI:** 10.3390/ma16010072

**Published:** 2022-12-21

**Authors:** Galal Elsamak, Ali Abdullah, Magdy I. Salama, Jong Wan Hu, Mahmoud A. El-Mandouh

**Affiliations:** 1Department of Civil Engineering, Faculty of Engineering, Kafrelsheikh University, Kafrelsheikh 33511, Egypt; 2Department of Civil and Environmental Engineering, Incheon National University, Incheon 22012, Republic of Korea; 3Incheon Disaster Prevention Research Center, Incheon National University, Incheon 22012, Republic of Korea; 4Civil Construction Technology Department, Faculty of Technology and Education, Beni-Suef University, Beni-Suef 62511, Egypt

**Keywords:** normal-strength concrete, high-strength concrete, strain-hardening cementitious composite concrete, ultra-high-performance fiber concrete, punching, steel tubes, high-strength bolts

## Abstract

The punching shear failure of reinforced concrete (RC) flat slabs is an undesirable type of failure, as it is sudden and brittle. This paper presents an experimental and numerical study to explore the behavior of flat slabs made of different types of concrete under the influence of punching shear. Experimental tests were carried out on four groups of flat slabs, each group representing a different type of concrete: ordinary normal concrete (NC), high-strength concrete (HSC), strain-hardening cementitious composite concrete (SHCC), and ultra-high-performance fiber concrete (UHPFC). Each group consisted of six slabs, one representing an unreinforced control slab other than the reinforcement of the bottom mesh, and the others representing slabs internally reinforced with SHCC-filled steel tubes and high-strength bolts. An analytical equation was used to predict the punching shear capacity of slabs internally reinforced using steel assemblies. A numerical model was proposed using the ABAQUS program, and was validated by comparing its results with our experimental results. Finally, a case study was performed on large-scale slabs. The results showed that using steel assemblies inside NC slabs increased the slab’s punching shear capacity but does not completely prevent punching shear failure. Internally unreinforced slabs made of UHPFC and SHCC were able to avoid punching shear failure and collapse in a ductile bending pattern due to the high compressive and tensile strength of these types of concrete. The proposed analytical method succeeded in predicting the collapse load of slabs reinforced with steel assemblies with a difference not exceeding 9%.

## 1. Introduction

Flat slabs, when compared to solid slabs, represent the simplicity of the wood formwork, short implementation time, and good architectural form. Therefore, flat slabs are widely used in garages, halls, and shopping centers. One of the main disadvantages of flat slabs is that they are subject to high punching shear stresses over columns, but these can be avoided using traditional methods such as increasing the thickness of the slab, using column heads, or using drop panels. Several researchers have studied the punching shear behavior of unreinforced and reinforced flat slabs [1,2,3,4,5,6,7,8,9,10,11,12,13,14,15,16,17,18,19,20,21,22,23,24]. Harajli et al. [25] studied the punching shear behavior of slabs by inserting steel bolts into reserved holes inside the slab around the columns and then pre-stressing them; this study showed that the use of this technique was able to increase the slab’s ability to resist punching shear, increase the ductility, and change the failure pattern into a ductile bending collapse. In similar studies by Adetifa and Polak [26] and Baig et al. [27], the test results indicated that the use of shear bolts in the connection area increased the strength of the connection and significantly improved the ductility. Saleh et al. [28] presented an experimental and numerical study to explore the punching shear strengthening of RC flat slabs using post-installed steel bolts; from this study it was found that the proposed strengthening techniques were able to change the collapse pattern of slabs from shear to flexural and increased the load capacity and ductility.

Gomes and Regan [29] carried out an experimental study to explore the punching shear behavior of RC slabs with embedded I-shaped steel beams that acted as shear-heads; from this study, they found that all slabs failed by punching shear and slabs which used shear-heads gave a greater load. Bompa and Elghazouli [30,31], Ngekpe et al. [32], and Zhou et al. [33] studied the behavior of flat slabs internally reinforced using different shapes of shear-heads and they found that the failure modes of slab-column connections embedded with steel skeletons were changed from punching shear to bending-punching, and that the slab column connections embedded with steel skeletons have higher punching shear capacity and structural ductility.

Afefy et al. [34] studied the punching shear behavior of RC slabs strengthened with UHPFC and found that by adding a thin layer of UHPFC on the tensile side, the bending performance of the slab was improved as it showed better crack distribution in the tensile side. However, the punching load slightly increased by about 5%. On the other hand, adding a UHPFC layer on the compression side enhanced the flexural and punching behavior of the slab. A new method has been presented to study the punching shear strength of UHPFC slabs by Hassan et al. [35].

There are several techniques for reinforcing concrete slabs to resist punching shear failures such as studs, headed bars, and closed stirrups. These techniques provide confinement of the punching shear failure surface and control crack propagation. Among the disadvantages of using these techniques are the difficulty of implementing them, the trouble to ensure that they are installed in the required places, as well as the limitation of their use in the case of slabs of small thickness due to the possibility of slipping, the possibility of the formation of the punching shear crack between its rows, and sometimes obstructing the path of the reinforcing steel of columns.

From the authors’ point of view and by searching in previous studies, it was found that there is a deficiency in studying the behavior of internally strengthened slabs to resist punching shear using high-strength steel bolts or using SHCC-filled steel-tubed sections, so this research aimed to experimentally and numerically [36] study the punching shear behavior of slabs constructed from NC, HSC, SHCC, and UHPFC as well as performing a case study on full-scale flat slabs. In addition, an analytical equation was used to predict the punching shear capacity of slabs internally reinforced using steel assemblies.

## 2. Experimental Program

### 2.1. General Description

To study the punching shear behavior of RC slabs, twenty-four concrete one-fifth scale slabs with dimensions 500 mm × 500 mm × 80 mm were designed and prepared, reinforced with a bottom steel mesh in both directions with four bars, 10 mm in diameter, and constructed from different types of concrete [33]. Experimental tests were carried out on four groups of flat slabs, each group representing a different type of concrete: normal strength concrete (NC), high-strength concrete (HSC), strain-hardening cementitious composite concrete (SHCC), and ultra-high-performance fiber concrete (UHPFC). Each group consisted of six slabs as follows: The first slab was a control slab that was not internally strengthened by any steel assemblies. The second slab was named RA attributed to being Radial Anchored and it was strengthened with a steel collar with a diameter of 160 mm made of a 12 mm circular bar and eight steel high-strength bolts, 16 mm in diameter and 50 mm in length, that were welded to this collar. The third slab was named PA attributed to being Plus Anchored and was strengthened with two steel skewers welded to form a plus; each skewer was 250 mm in length, 25 mm wide, and 3.5 mm thick, and two high strength bolts, 16 mm in diameter and 50 mm in height, were welded to the end of each skewer. The fourth slab was internally strengthened with the same technique as slab PA, but the skewer used was an octagonal shape and was named OA. The fifth slab was named PT attributed to its being Plus Tubed, as it consists of two perpendicular steel tubes, 250 mm in length in both directions. The sixth slab was named OT attributed to being Octagonally Tubed, and was internally strengthened with eight octagonal tubes confined to a circle with a diameter of 250 mm. All the tubes used in this study have dimensions of 40 mm × 40 mm × 2 mm and were filled with SHCC 35 days before the casting of slabs. One of the objectives of this paper was to use steel tubes as internal reinforcements to increase the ability of the reinforced concrete slabs to resist the punching shear. Since it is difficult to overcome the occurrence of honeycombing during the pouring of the concrete for these slabs, there was a need to fill these steel tubes before pouring the concrete that has excellent tensile and compressive performance and does not contain large aggregates (i.e., SHCC). The concrete cover was 5 mm for all specimens. Figure 1 and Figure 2 illustrate the strengthening techniques used. Table 1 illustrates the basic information of the tested specimens. A strain gauge was fixed to the middle bottom steel mesh of the slabs nearest to the slab center to track the strain on the steel bar during loading.

It is worth noting that we used SHCC-filled steel tubes to resist the punching load through tension in steel tubes (Vs) and shearing in SHCC (Vc) (see Figure 3 as shown below). This is because this type of concrete has good tensile, compressive, and shear behavior. In addition, these steel tubes were filled before pouring so that no honeycombing would occur inside them.

### 2.2. Material Properties

For the preparation of the required types of concrete, we used Ordinary Portland cement (Type I), sand, and crushed basalt as aggregate. The maximum aggregate size was 10 mm, and was added to free mixing water, fly ash, silica fume, steel fibers, polypropylene fibers, and superplasticizer. The quantities of the components in kilograms used per cubic meter to prepare each type of concrete are shown in Table 2. Wood formwork and casting work are shown in Figure 4. Curing was carried out for slabs by spraying them with water for 28 days. When casting any slab, three cylinders, 150 mm in diameter × 300 mm in height, were cast from each type of concrete to be tested under compression on the day of slab testing to obtain the concrete compressive strength (*f_c_’*). Compression tests were carried out for the concrete cylinders on the day of slab testing and the average compressive strength was 41, 65, 87, and 133 N/mm^2^ for NC, HSC, SHCC, and UHPFC, respectively. Direct tensile tests were also carried out for samples of steel reinforcement and steel tubes used to reinforce slabs, and from these tests, it was found that the yield stress of the bottom reinforcement meshes and tubes were 413 N/mm^2^ and 290 N/mm^2^, respectively. From the data sheet attached to the bolts, it was found that their yield stress is 900 N/mm^2^.

For the preparation of the SHCC, we used polypropylene fibers with a length of 12 mm and a diameter of 0.012 mm with a tensile strength of 400 MPa and an elongation of 80%. For the preparation of the UHPFC, a mixture of straight- and hooked-end steel fibers was used in a ratio of 1:1 by weight. The straight- and hooked-end steel fibers had a diameter of 0.2 mm and 0.35 mm, a length of 13 mm and 25 mm, and an aspect ratio of 65 and 71 with a tensile strength of 2500 MPa and 2550 MPa, respectively. Direct tensile tests were performed on SHCC and UHPFC specimens in the same manner as that of Zeng et al. [37]. The geometry of the samples and tensile stress–strain curves are shown in Figure 5, and it is clear that the tensile strength was 5.7 MPa and 6.9 MPa for SHCC and UHPFC, respectively.

## 3. Test Setup and Instrumentation

The slabs were placed horizontally on a steel support consisting of 4 sides from I-shaped beams and welded together to form a square, and solid steel bars were welded parallel to the webs to the upper flanges; the distance between the center of the opposite bars is 450 mm. The aforementioned support was installed above the bearings inside the loading frame, and the slab was placed on top so that the center of these bearings, the support, and the slab were below the loading cylinder that was connected to a load cell. To convey the load from the loading piston to the slab, hollow steel cylinders with an outer diameter of 90 mm and an inner diameter of 70 mm were used. The cylinders were welded to a steel cap of 10 mm thickness at each end. The cylinders were stacked on top of each other to form a column with a total length of 840 mm. We did not observe any lateral deviation for this column resulting from buckling or deformations during loading, and a nonlinear buckling analysis was performed for this column alone, and it was found that the buckling critical load was 5700 kN, which is much greater than the target load for the collapse of all samples. An LVDT was installed under the slab in its center to measure the vertical displacement with loading. All data from the load cell, LVDT, and strain gauge were fed to a data logger that was connected to a computer. All tests were carried out at the RC lab, in the Faculty of Engineering, Kafrelsheikh University. Figure 6 shows the test setup.

## 4. Results and Discussion

### 4.1. For NC and HSC Groups

By tracking cracks, it was found that cracks began to appear on the bottom side of the slab at approximately 16% and 19% of the maximum load for each of the NC and HSC slabs, respectively. With the increase in loading, the cracks increased, and the width of the existing cracks increased. With the increase in loading, more diagonal cracks appeared and began to spread toward the corners of the slab. When approaching the end of loading, radial cracks occurred on the top surface of the slab around the loading plate, then the concrete began to collapse in compression, followed by a cracking sound. Figure 7, Figure 8, Figure 9 and Figure 10 show the failure patterns for all tested slabs.

The behavior of the load–displacement curves of the NC and HSC slabs was similar and occurred in two stages. The first stage was the linear stage and started from the beginning of loading until the occurrence of the first crack in the slabs. The second stage was the non-linear stage and started after the first stage until the maximum load was reached. Figure 10 shows the load–displacement curves for all slabs and Table 3 gives a summary of all experimental results. From Figure 7, Figure 8, Figure 9 and Figure 10 and Table 3, it should be noted that the slabs in the NC group collapsed with the punching shear in a brittle pattern, and no reinforcing technique succeeded in preventing this. The normalized punching load for the control slab complies with ACI 318-14 [8] where it was stipulated that the normalized punching load at the critical section should not be more than 0.33. The normalized punching load is defined as (P_u_/(√ *f_c_’* × b_0_ × d)) where (P_u_) is the slab’s maximum load, *f_c_’* is the concrete’s compressive strength, (b_0_) is the circumference of the collapsing section at a distance (d/2) from the face of the loading surface, and (d) is the effective depth of the slab. All reinforcing techniques succeeded in increasing the slab’s punching shear load as it increased by 13%, 18%, 29%, 28%, and 37% for RA, PA, OA, PT, and OT techniques, respectively. The OT technique gave the largest increase in the punching shear load, reaching 37%.

The slabs in the HSC group exhibit very similar behaviors to those in the NC group except that they have a greater punching shear load value due to the high compressive strength of HSC compared to NC. From observation of the normalized punching load for the HSC group compared with the NC group, it was assumed that the punching shear load is proportional to the square root of the concrete compressive strength *f_c_’*. It must be emphasized that no yielding was observed in any of the slabs in the NC group or HSC group.

### 4.2. SHCC and UHPFC Groups

For SHCC and UHPFC slabs, bending cracks appeared on the bottom surface of the slab at a load of about 6% of the maximum load. With increasing load, the number of cracks increased, and the width of the existing cracks increased. When nearing the maximum load, radial cracks on the top surface of the slab were observed around the loading plate, and compression collapses of the concrete occurred. Afterwards, increasing the load resulted in a rapid increase in the displacement along with the stability of the load value.

It is clear from Figure 11 that the behavior of the load–displacement curves of SHCC and UHPFC slabs are similar and pass through three stages. The linear stage started from the beginning of loading until yielding occurred in the lower reinforcement. The second stage was non-linear and started after the first stage until the maximum load was reached. The third stage was the load stabilization stage with significant increases in displacement values.

Control slabs made of SHCC or UHPFC successfully avoided brittle punching shear failure and collapsed by bending in a ductile collapse pattern. It was observed that the reinforcement yield occurred by calculating the ductility as the ratio between the displacement at the maximum load (Δ_u_) to the displacement when the yield occurs (Δ_y_). For slabs made of SHCC, the PT technique gave the largest ductility of 2.25, while the OT technique gave the largest increase in the ultimate load of the slab of 40%. For slabs made of UHPFC, the OT technique gave the largest ductility of 11.26, while the OA technique gave the largest increase in slab ultimate load of 35%; this difference is due to the better tensile and compressive strength of UHPFC when compared to SHCC. From Figure 10 the dominant failure pattern on the control unreinforced slab is clearly punching shear failure, and the use of steel assemblies succeeds in moving the collapsed section from the vicinity of the column (loading) to the edge of the end of the steel assemblies. Figure 12 shows the ultimate loads for the different tested slabs.

As shown in Figure 9 and Figure 10, the cracks in slabs made from SHCC or UHPFC are large in number and their width is small compared to slabs made from NC or SHCC, and these cracks extend over almost the entire surface of the slab. In addition, Figure 10 clearly shows that there is a gradual increase in the stiffness (the slope of the linear portion of the load–displacement curve) for NC, HSC, SHCC, and UHPFC.

## 5. Numerical Modeling

To obtain a numerical model capable of modeling ordinary RC slabs or slabs reinforced with steel assemblies to resist punching shear, the ABAQUS program [36] was used, which is one of the most famous structural analysis programs that is based on the finite element (FE) method. Nonlinear analysis is available in the ABAQUS software, which can take into account the nonlinearity of the material or the nonlinearity of the geometry. To ensure the success of the proposed numerical model, verification was performed by experimentally testing the RC slabs. The C3D10 element (a 10-node quadratic tetrahedron element) was used to model the circumferential steel support under the slab and for the in-slab steel assemblies used to resist punching shear, while the element C3D8R (an 8-node linear brick, reduced integration, hourglass control element) was used to model the concrete slabs and the loading plate on top of the slab. The element T3D2 (a 2-node linear 3-D truss) was used to model the steel bars used to reinforce the slabs. To save analysis time, only a quarter of the slab was modeled, depending on the characteristics of their geometric symmetry. The interaction between the slab and the circumferential steel support below it was considered a hard contact interaction that allow separation without any friction properties. The circumferential steel support was fixed. Figure 13, Figure 14, Figure 15, Figure 16 and Figure 17 show the numerical models for different RC slabs and steel skeletons. For material modeling, a CDP (concrete damaged plasticity) model was used for modeling concrete, while the elastic-perfect plastic behavior was used to model the steel elements. Table 4 shows the elastic and plastic parameters used for materials, and the behaviors of different materials used in the numerical modeling are shown in Figure 18. The Carreira and Chu [38] constitutive model was used to constructing stress–strain curves for NC and HSC, while the model of Zhou et al. [39] was used to construct stress–strain curves for SHCC and UHPFC. The elastic-perfect plastic behavior was used to model the steel elements.

To choose the appropriate mesh size, a mesh sensitivity analysis was performed on the NC specimens using three different sizes: 30 mm × 30 mm, 20 mm × 20 mm, and 10 mm ×10 mm. Figure 19 shows the resulting load–displacement curves, which shows that the best simulation was produced using the size 10 × 10 mm, and accordingly, this mesh size was used to model all slabs. Figure 20 shows a comparison of the numerical and experimental load–displacement curves and Figure 21, Figure 22, Figure 23 and Figure 24 show the numerical failure patterns. The results showed that there is a strong agreement between the results of the numerical analysis with the experimental results in the linear and non-linear stages. Table 5 gives a comparison of experimental and numerical results for maximum load and maximum displacement. It should be noted that the tolerances in the maximum load and maximum displacement did not exceed 5% and 25%, respectively.

The numerical modeling used to simulate the behavior of slabs made of NC, HSC, SHCC, and UHPFC, reinforced by steel assemblies in the high-stress region of the punching shear, was successful, which confirms the compatibility of the materials models, elements used, and the interaction method. The effect of the beneficial properties of SHCC in tension or compression was observed in its stress–strain curves, while the high value of its dilation angle indicated its benefits in resisting shearing. The finite element method using the ABAQUS program was an effective method for analyzing the behavior of flat slabs.

## 6. Case Study

After confirming the success of the numerical modeling, the numerical study was expanded to explore the behavior of slabs at full scale using two slabs. The first control slab was named Slab B0. The second slab was similar to Slab B0, but was reinforced using the PT technique and named Slab B-PT. The horizontal plane shown in Figure 25a shows the details of these slabs. They are slabs from two spans in both directions. The span length between the column’s center lines was 6000 mm. The thickness of the slabs was 200 mm, reinforced with an upper and lower mesh of 7Φ18/m in both directions, and rested on columns with 300 mm × 300 mm cross sections that were reinforced with four bars with a 18 mm diameter. The concrete used for the slabs was NC. The details of the reinforcing for Slab B-PT are shown in Figure 25b. The NC, SHCC, reinforcements, and steel used have the same characteristics as previously mentioned in the numerical verification study. This study aimed to explore the behavior of these slabs under the influence of vertical loads. As in the previous experiment, a quarter of the slab was modeled and subjected to a uniform distributed load W_u_ (kN/m^2^) on its surface as shown in Figure 26. Figure 27 shows the load–displacement curves for the analyzed slabs. The displacement was monitored at the mid-point of the edge bay. Slab B0 collapsed by punching shear in a brittle pattern, while Slab B-PT collapsed by flexural in a ductile pattern. The PT technique was able to increase the ultimate load capacity of the slab by 61.3%, as the ultimate load of the slab increased from 23.25 kN/m^2^ to 37.5 kN/m^2^ for the two slabs, B0 and B-PT, respectively. Figure 28 shows the failure patterns of the analyzed slabs.

## 7. Prediction of Punching Shear Load

This section aimed to reach an analytical method to predict the punching shear capacity of control slabs or internally reinforced slabs using steel assemblies. The total punching shear load of concrete slabs or internally reinforced using steel assemblies (V_u_) can be considered as the sum of the two punching shearing loads: the punching shear load of the concrete (V_uc_) and the punching shear load of the internal reinforcement used (V_us_).
V_u_ = V_uc_ + V_us_(1)

The punching shear load resisted by the concrete can be calculated from Equation (2) where t_cf_ is the strength of the concrete to resist punching shear; b_0_ is the perimeter of the punching critical section which can be considered in the case of control slabs at a distance of d/2 from the column face; and d is the effective depth of the slab.
V_uc_ = t_cf_ b_o_ d(2)

According to Muttoni and Fernández [40], τ_cf_ can be calculated from Equation (3) where $f_{c}^{\prime }\;$is the compressive strength of the concrete in N/mm^2^; *d_g_* is the maximum size of the aggregate; *d*_*g*0_ is 16 mm; and β is the rotation of slab.
(3)$$\tau_{\mathrm{cf}}=\frac{\;0.75\;{\left(f_{c}^{\prime }\right)}^{0.5}}{1+15\;\left(\frac{\beta d}{d_{g0}+d_{g}}\right)}$$

β can be calculated from Equation (4) where $r_{s}$ is the distance between the lines of contraflexure and the column, *t* is the total thickness of the slab; $f_{y}$ is the yield stress of the bending reinforcement; $E_{s}$ is the modulus of elasticity of the bending reinforcement; $M_{s}$ is the bending moment acting at the top of the column; and $M_{r}$ is the moment of resistance of the slab which can be calculated from Equation (5) according to ACI [8], where ρ is the bending reinforcement ratio.
(4)$$\beta =1.5\;\frac{r_{s}}{t}\;\frac{f_{y}}{E_{s}}{\left(\frac{M_{s}}{M_{r}}\right)}^{1.5}$$
(5)$$M_{r}=\rho \;f_{y}d^{2}\;(1\text{–}0.59\;\rho \;\frac{f_{y}}{f_{c}^{\prime }}\;)$$

According to Timoshenko and Woinkowsky-Krieger [41], the bending moment acting on the slab can be calculated from Equation (6), where L is the slab length.
*M_s_* = V_u_ L/14(6)

The capacity of the internal reinforcement to resist punching shear can be calculated from Equation (7) where σ_s_ is the steel stress in the vertical direction; A_s_ is the area of the bolt used or the sum of the areas of webs that existed in a distance d in the direction of the horizontal axis of the slab; α is the angle between the axis of the bolt or the web with the horizontal axis of the slab which will be considered to be 90 degrees in the current study.
V_us_ = σ_s_ A_s_ sinα(7)
(8)$$\sigma_{s}=\frac{E_{ss}\beta \;}{6}\;(\sin\alpha +\cos\alpha )\;(\sin\alpha +\frac{f_{bd}}{f_{ys}}\frac{d}{1.13\;{\left(A_{\mathrm{ss}}\right)}^{0.5}\;})f_{ys}$$
where $f_{ys}$ and $E_{ss}$ are the yield stress and modulus of elasticity of the steel used, respectively; $f_{bd\;}$is the bond stress between the steel and the concrete which can be considered as 25 N/mm^2^ for the embedded reinforcement; and $A_{\mathrm{ss}}$ is the area of the steel used to resist punching through a distance d on the horizontal axis of the slab. In the case of the ability of the internal shear reinforcement to avoid collapse by punching shear, the ability of the slab to collapse by bending (V_flex_) can be calculated by Equation (9) according to Elstner and Hognestad [42], where c is the diameter of the column.
$$V_{\mathrm{flex}}=8\;M_{r}(\frac{1}{1-\left(\frac{c}{L}\right)}-3+2\;\sqrt{2})$$

Table 5 gives the results of the analytical study for the experimentally tested slabs, which shows that the difference between the analytical and experimental punching load does not exceed 9%. The presented analytical method succeeded in predicting the collapse load, whether by shear or by bending.

## 8. Conclusions

This paper presented an experimental and numerical study to explore the behavior of flat slabs made of different types of concrete under the influence of punching shear and reinforced internally by high-strength steel bolts or SHCC-filled steel-tubed sections. From this study it was shown that:By using steel assemblies embedded in the NC slabs, the punching shear capacity of the slab was increased but punching shear failure was not avoided.Internally unreinforced slabs made of SHCC and UHPFC were able to avoid punching shear failure and collapse in a ductile bending pattern due to the high compressive and tensile strength of these types of concrete.In the case that the dominant failure pattern on the control unreinforced slab is punching shear failure, the use of steel assemblies succeeded in moving the collapsed section from the vicinity of the column (loading) to the edge of the end of the steel assemblies.For slabs made of SHCC, the PT technique gave the largest ductility of 2.25, while the OT technique gave the largest increase in the ultimate load of the slab of 40%. For slabs made of UHPFC, the OT technique gave the largest ductility of 11.26, while the OA technique gave the largest increase in the slab ultimate load of 35%; this difference is due to the better tensile and compressive strength of UHPFC compared to SHCC.The numerical modeling used to simulate the behavior of slabs made of NC, HSC, SHCC, and UHPFC, reinforced internally by steel assemblies in the high-stress region of the punching shear, was successful, which confirms the compatibility of the models of the materials used and the interaction method. The finite element method using the ABAQUS program is an effective method for analyzing the behavior of flat slabs.The proposed analytical method succeeded in predicting the collapse load of slabs reinforced with steel assemblies with a difference not exceeding 9%.

Future work: Studying the behavior of RC slabs made of different types of concrete and reinforced to resist punching shear using UHPFC-filled aluminum tubed sections.

## Figures and Tables

**Figure 1 materials-16-00072-f001:**
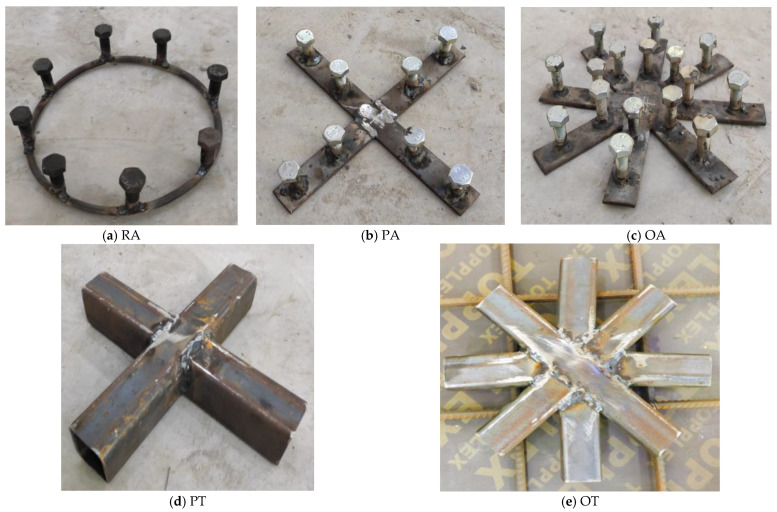
The steel assemblies used.

**Figure 2 materials-16-00072-f002:**
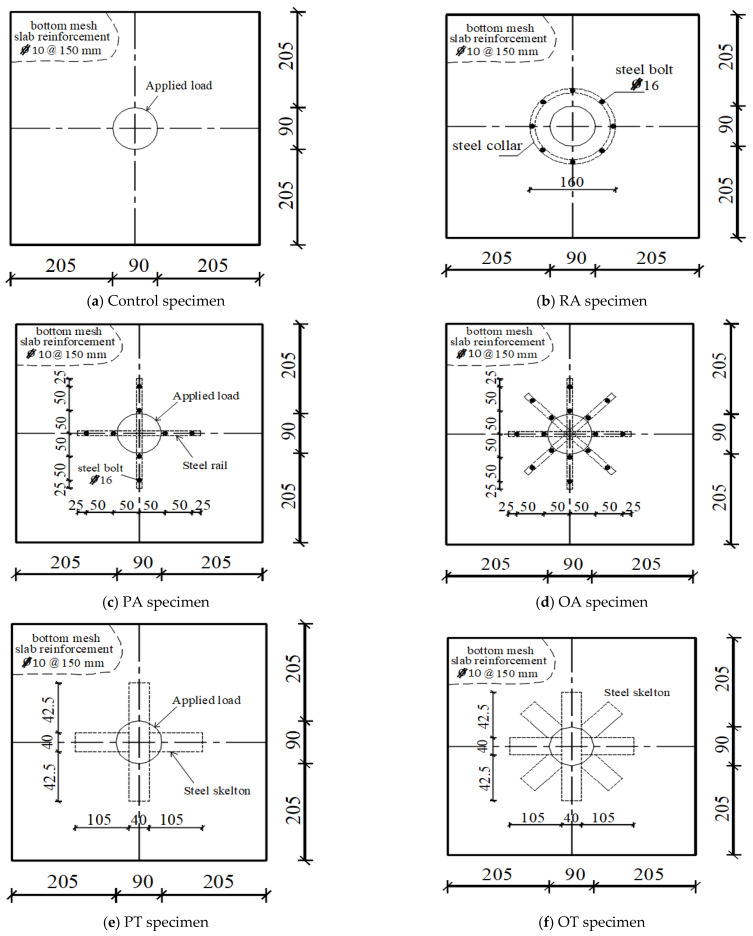
Dimensions and details of specimen configurations.

**Figure 3 materials-16-00072-f003:**
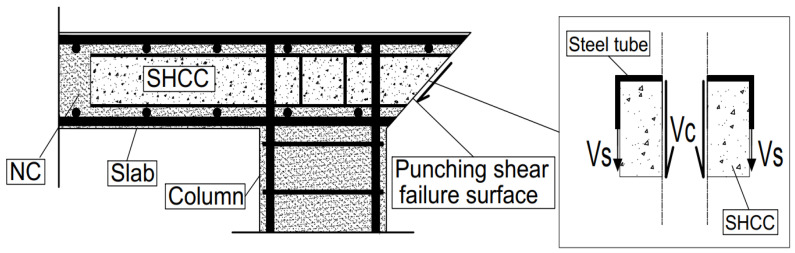
Distribution of shear forces in slabs reinforced with SHCC-filled steel tubes.

**Figure 4 materials-16-00072-f004:**
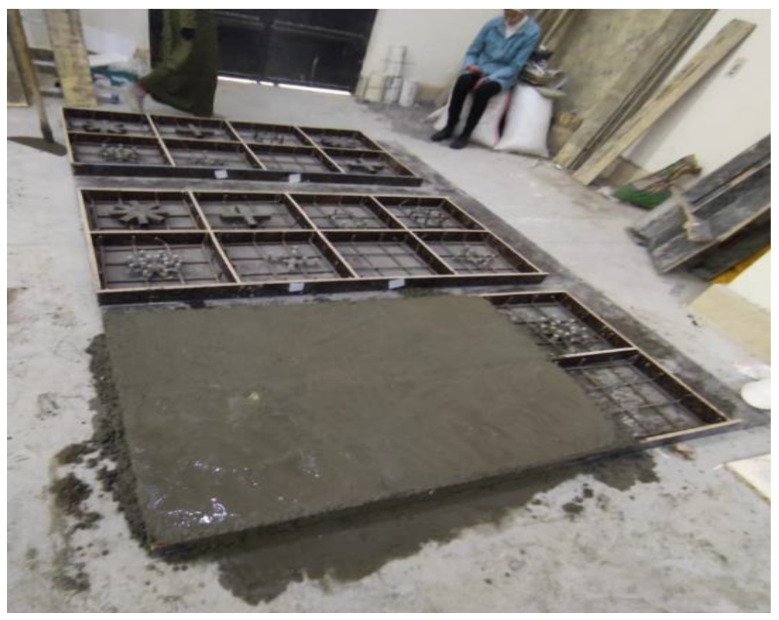
Wood formwork and casting work.

**Figure 5 materials-16-00072-f005:**
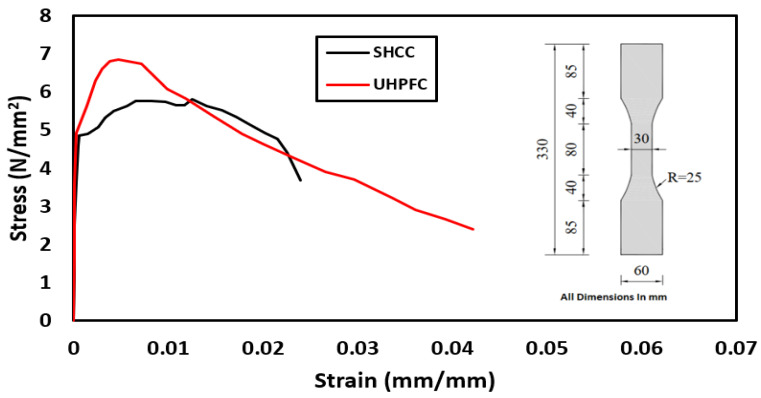
The geometry of the samples and tensile stress–strain curves.

**Figure 6 materials-16-00072-f006:**
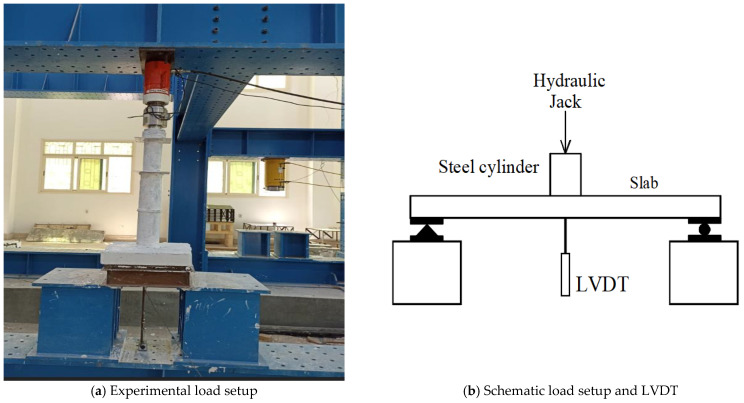
Test setup.

**Figure 7 materials-16-00072-f007:**
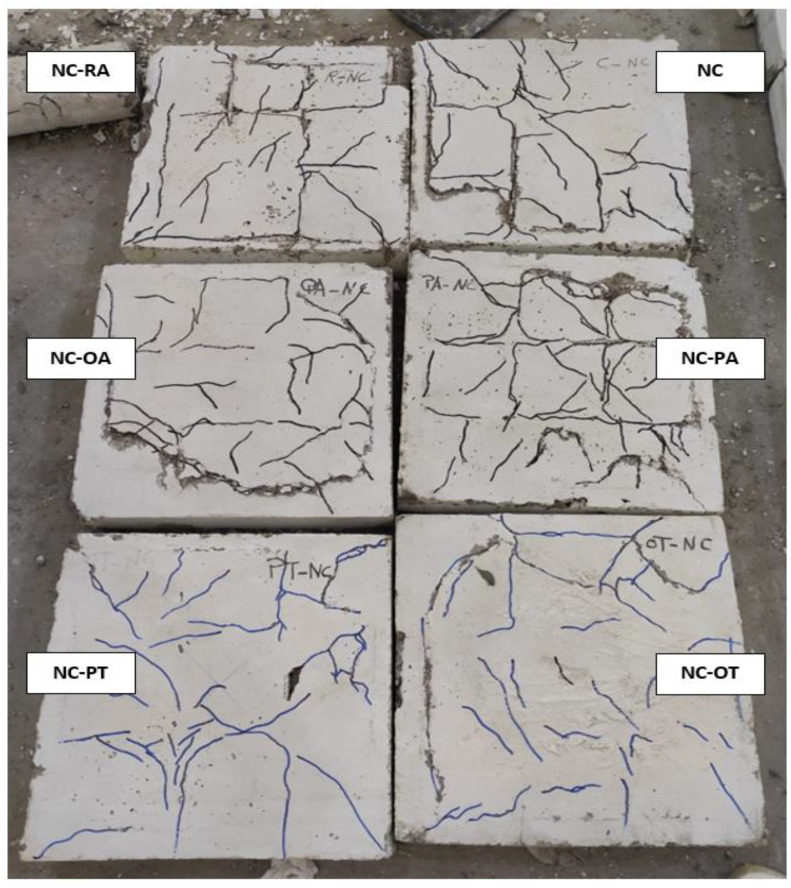
NC specimens cracks.

**Figure 8 materials-16-00072-f008:**
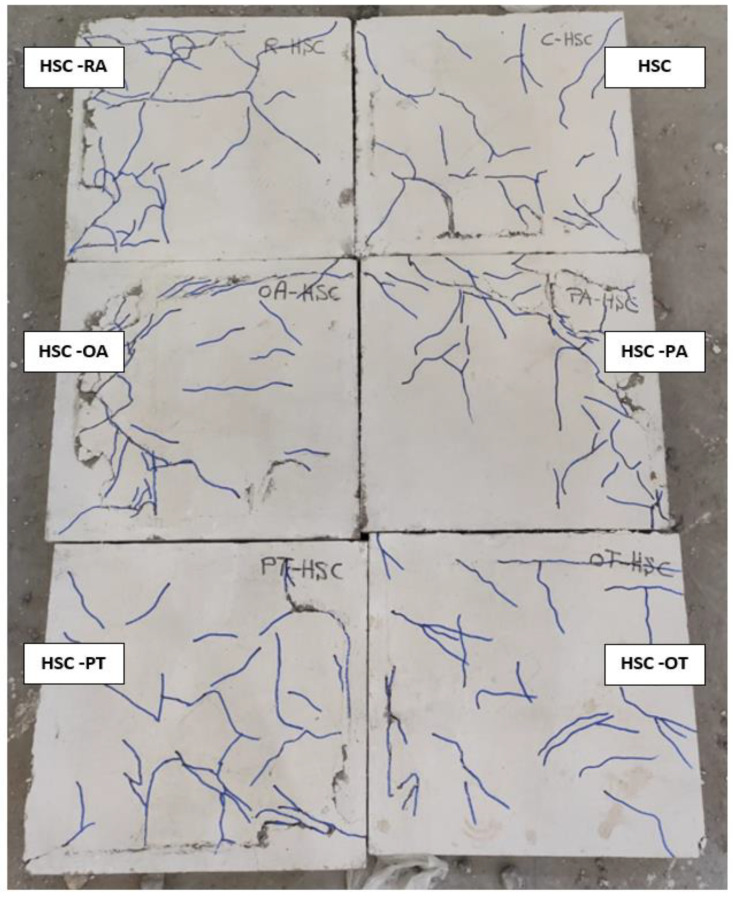
HSC specimens cracks.

**Figure 9 materials-16-00072-f009:**
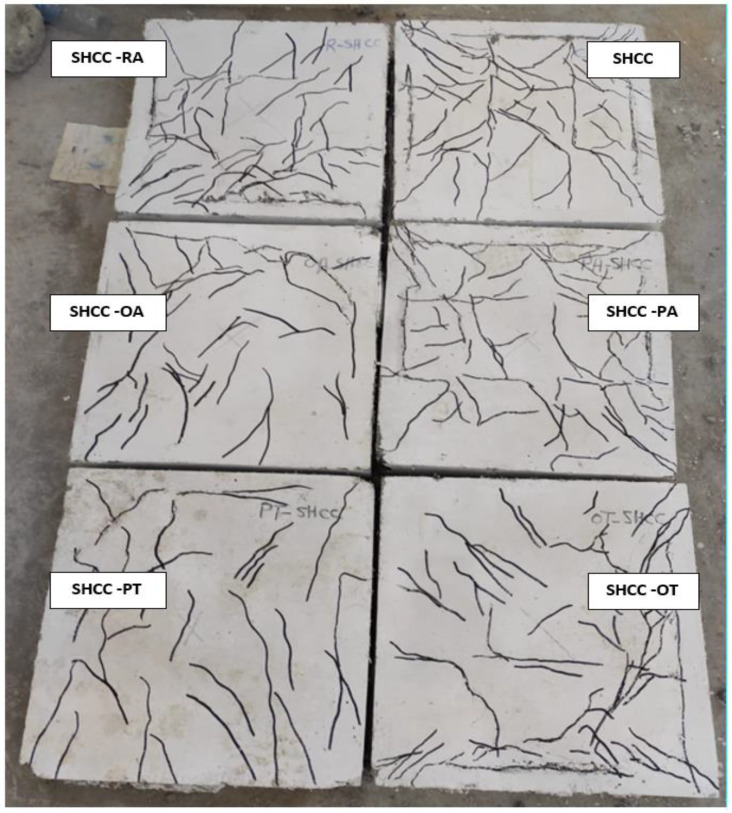
SHCC specimens cracks.

**Figure 10 materials-16-00072-f010:**
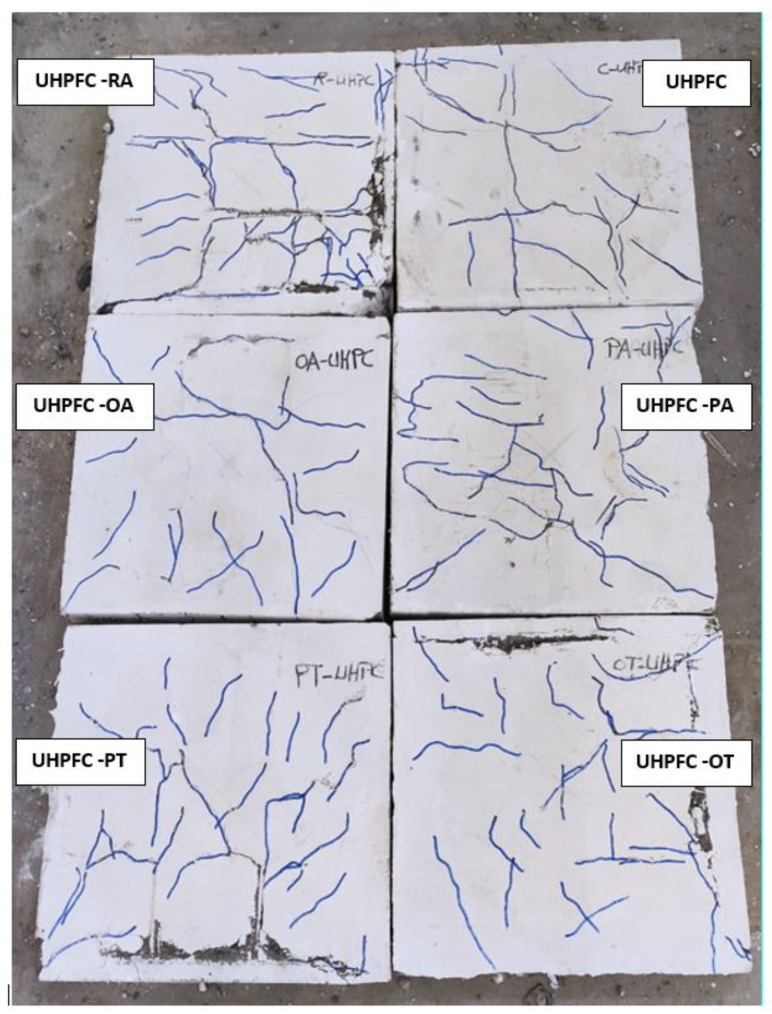
UHPFC specimens cracks.

**Figure 11 materials-16-00072-f011:**
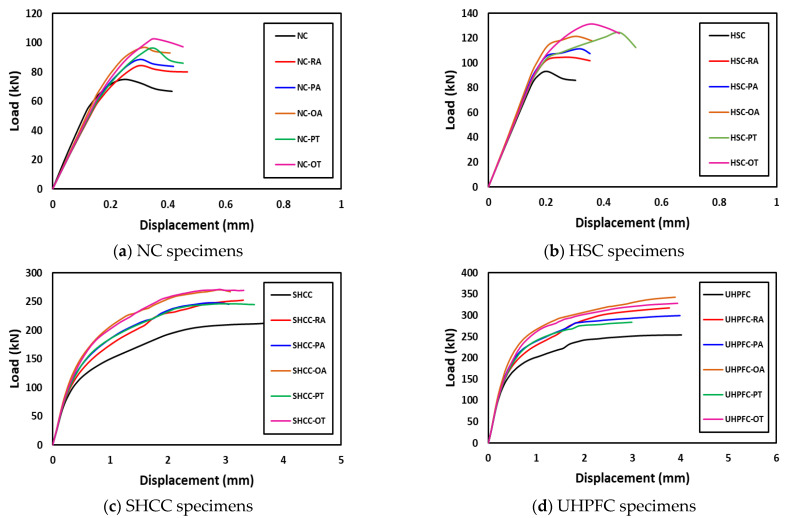
Load–displacement curves for tested specimens.

**Figure 12 materials-16-00072-f012:**
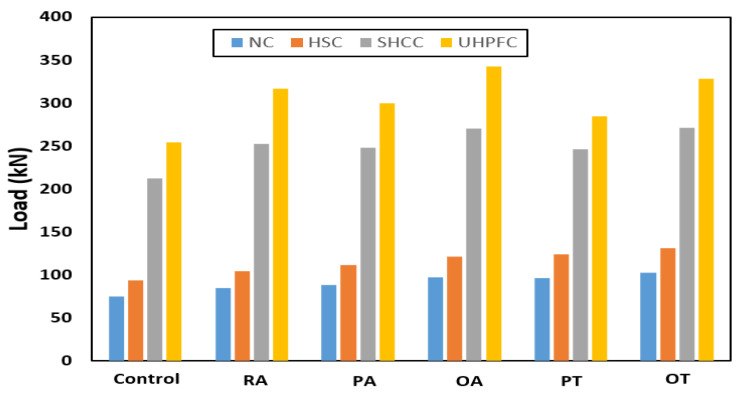
Ultimate loads for different tested specimens.

**Figure 13 materials-16-00072-f013:**
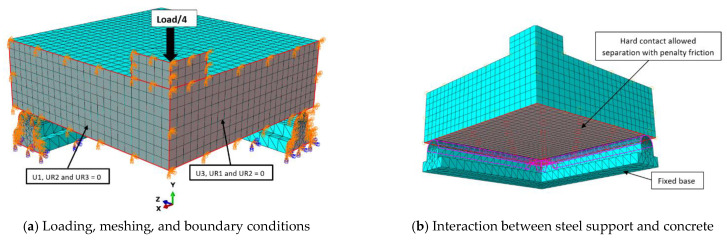
Numerical model of NC slab.

**Figure 14 materials-16-00072-f014:**
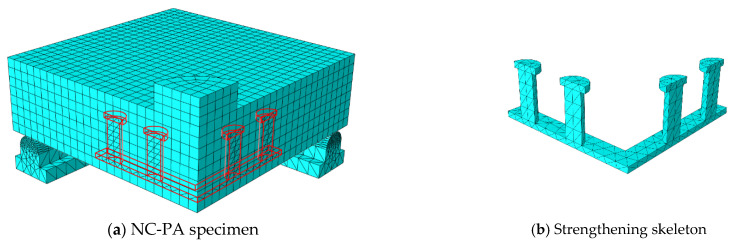
Model of NC-PA slab.

**Figure 15 materials-16-00072-f015:**
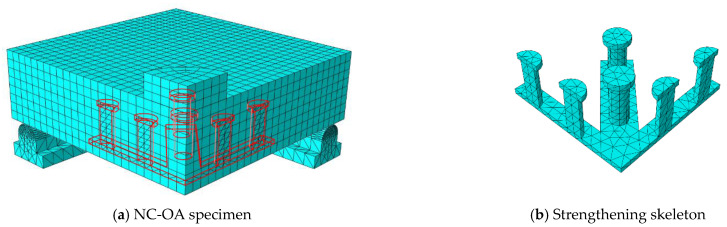
Model of NC-OA slab.

**Figure 16 materials-16-00072-f016:**
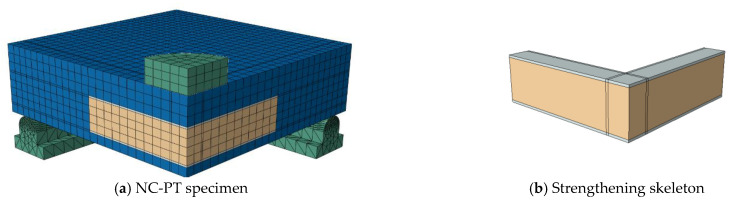
Model of NC-PT slab.

**Figure 17 materials-16-00072-f017:**
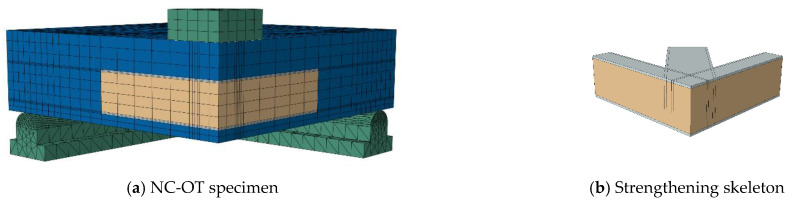
Model of NC-OT slab.

**Figure 18 materials-16-00072-f018:**
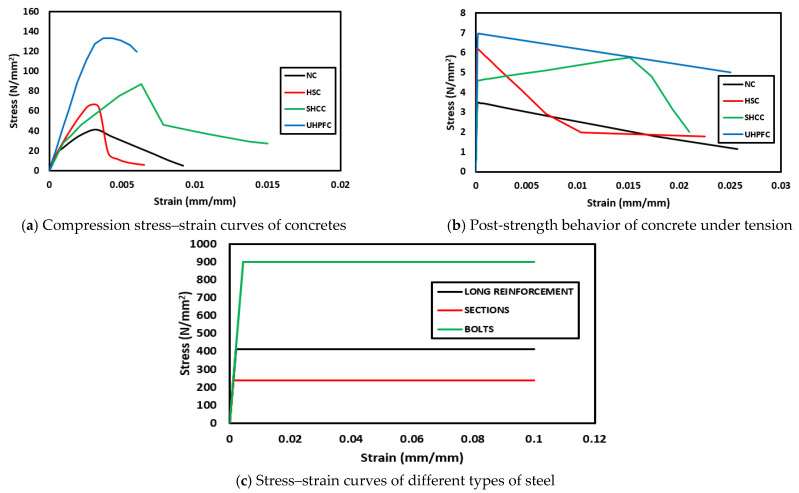
Behavior of materials used in the numerical modeling.

**Figure 19 materials-16-00072-f019:**
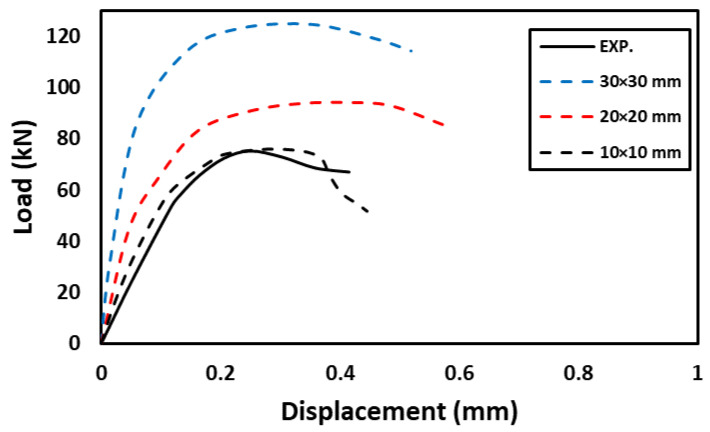
The effect of mesh size on the load–displacement curves of NC specimens.

**Figure 20 materials-16-00072-f020:**
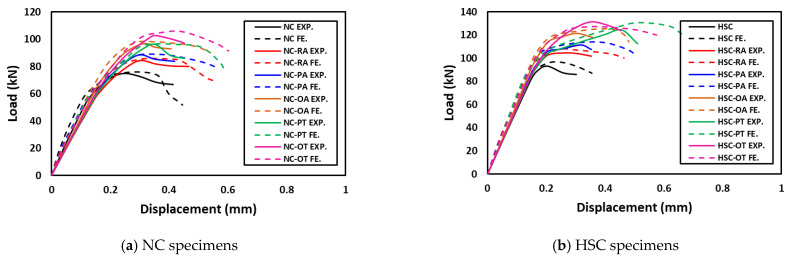
Numerical versus experimental load–displacement curves.

**Figure 21 materials-16-00072-f021:**
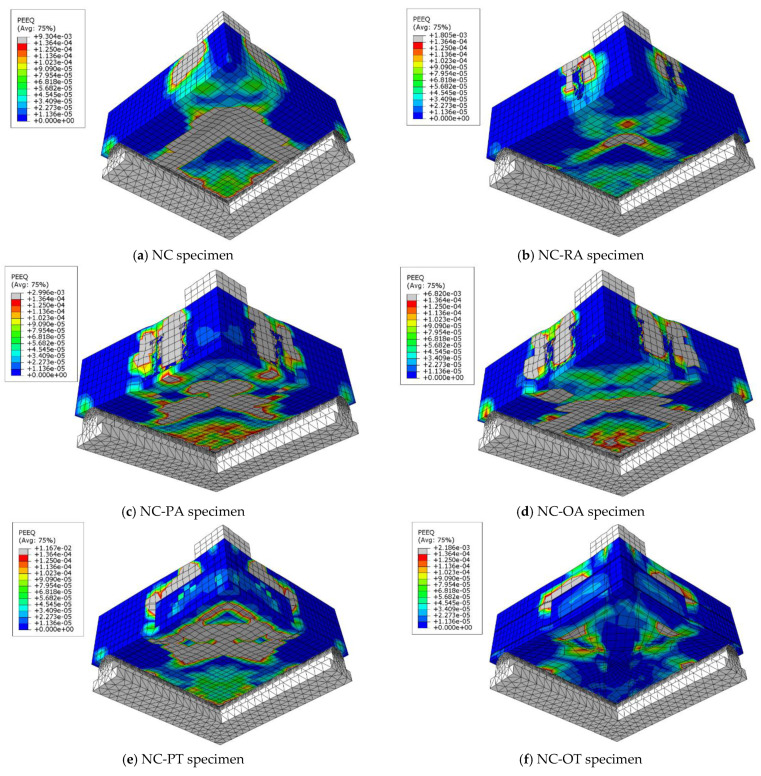
Analytical NC specimens cracks.

**Figure 22 materials-16-00072-f022:**
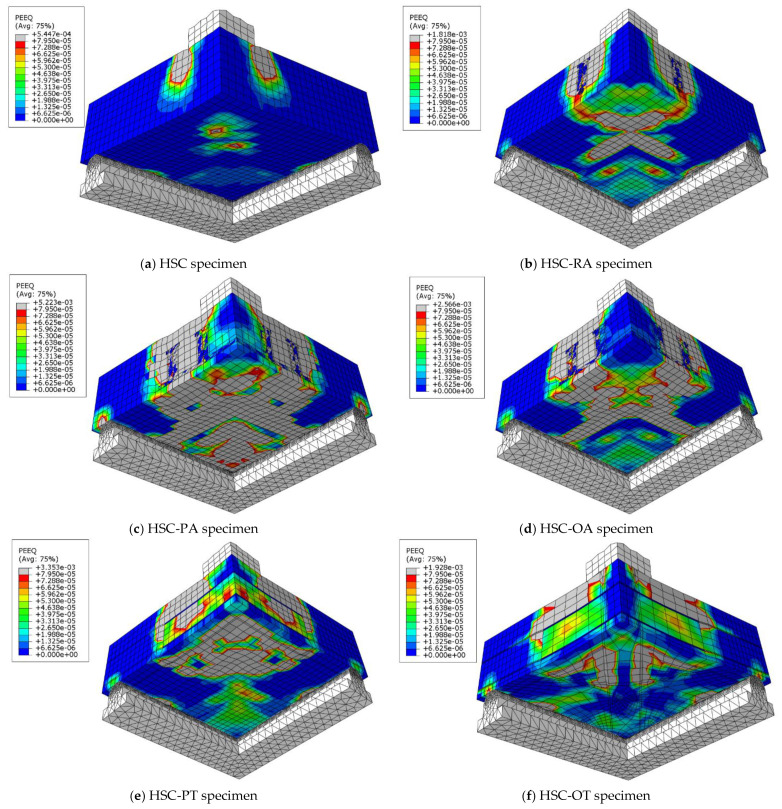
Analytical HSC specimens cracks.

**Figure 23 materials-16-00072-f023:**
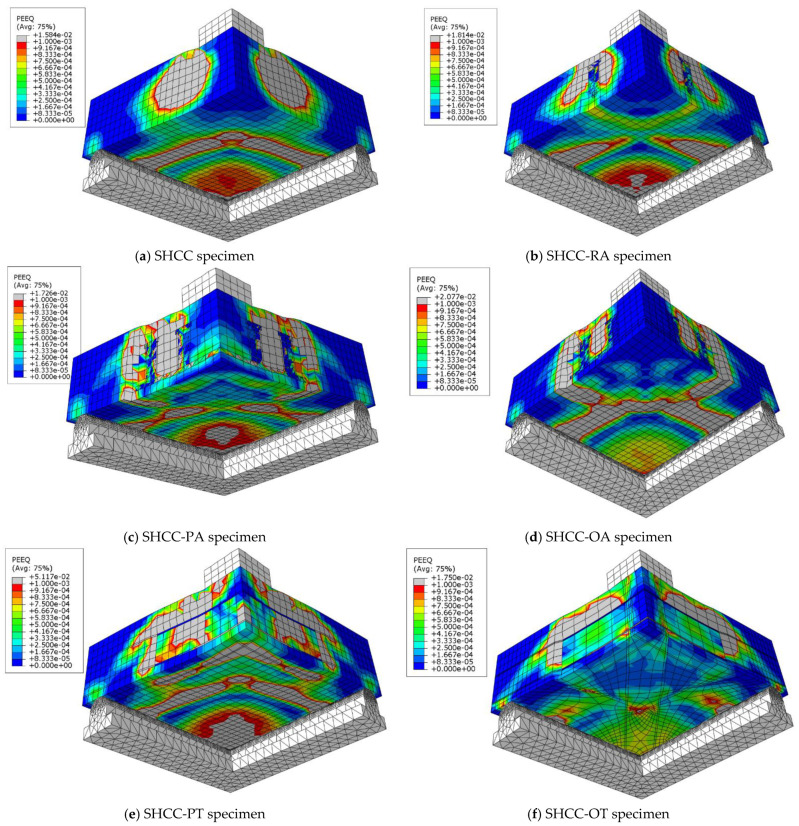
Analytical SHCC specimens cracks.

**Figure 24 materials-16-00072-f024:**
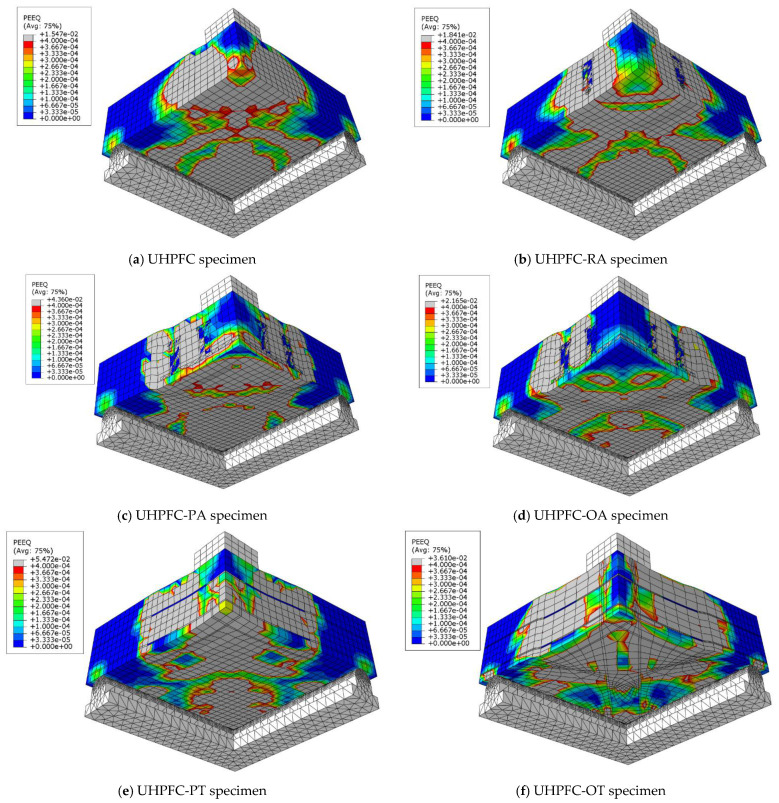
Analytical UHPFC specimens cracks.

**Figure 25 materials-16-00072-f025:**
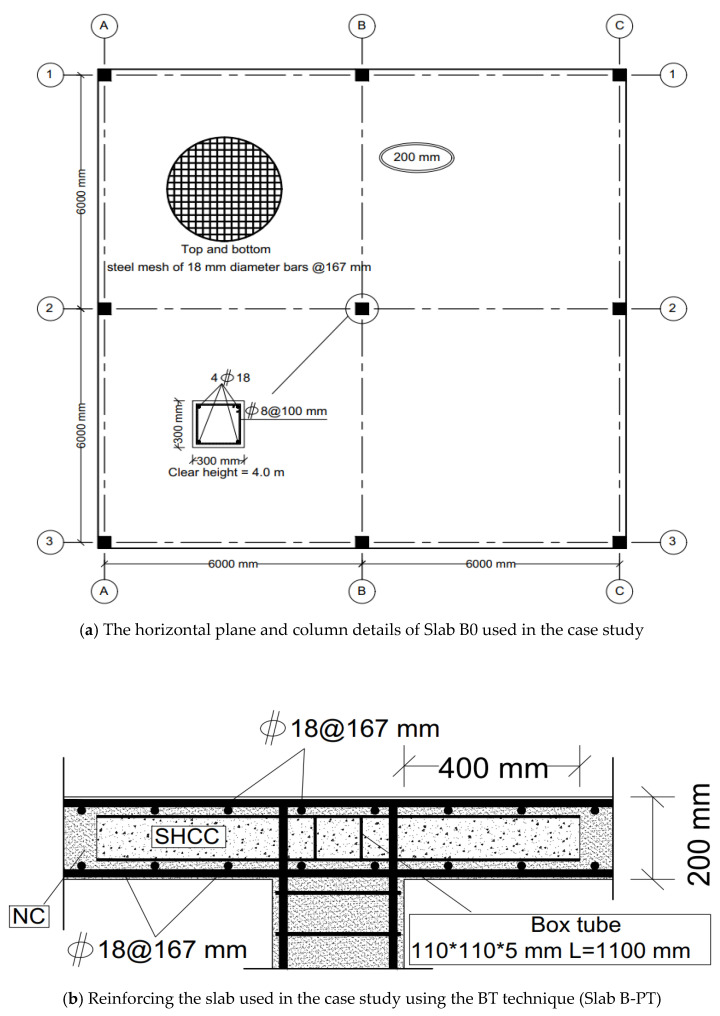
Details of the slabs used in the case study.

**Figure 26 materials-16-00072-f026:**
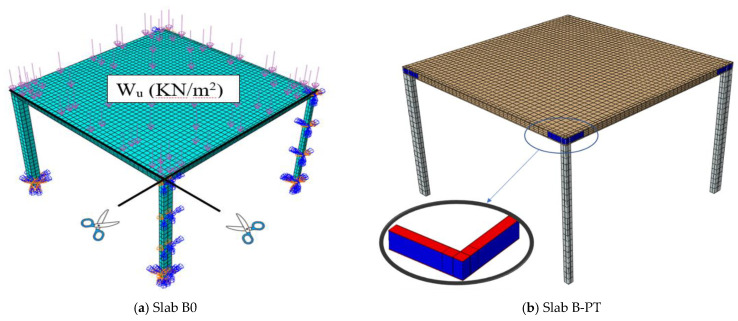
Numerical models of the slabs used in the case study.

**Figure 27 materials-16-00072-f027:**
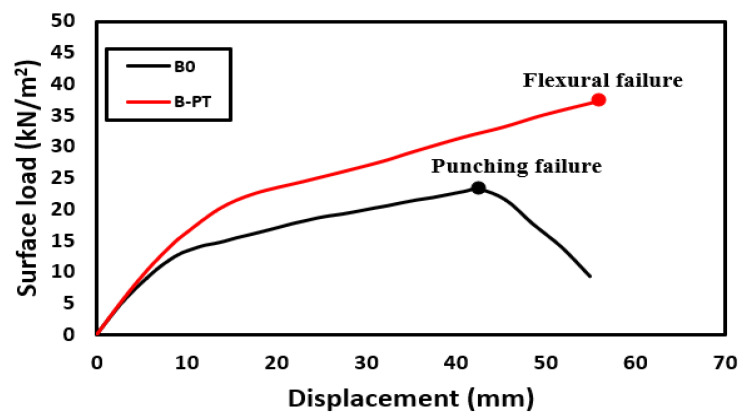
Load–displacement curves for slabs analyzed in the case study.

**Figure 28 materials-16-00072-f028:**
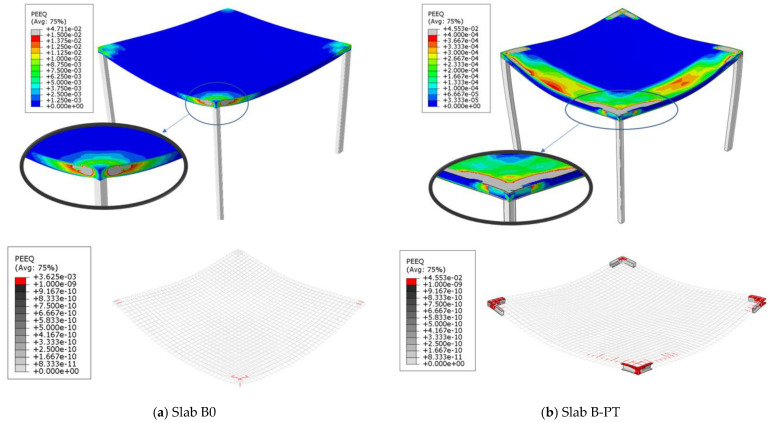
Numerical failure patterns for tested slabs.

**Table 1 materials-16-00072-t001:** Basic information of tested specimens.

Group	Slab ID	Concrete Grade	Type of Steel Skelton
1	NC	NC	-
	NC-RA	NC	RA
	NC-PA	NC	PA
	NC-OA	NC	OA
	NC-PT	NC	PT
	NC-OT	NC	OT
2	HSC	HSC	-
	HSC-RA	HSC	RA
	HSC-PA	HSC	PA
	HSC-OA	HSC	OA
	HSC-PT	HSC	PT
	HSC-OT	HSC	OT
3	SHCC	SHCC	-
	SHCC-RA	SHCC	RA
	SHCC-PA	SHCC	PA
	SHCC-OA	SHCC	OA
	SHCC-PT	SHCC	PT
	SHCC-OT	SHCC	OT
4	UHPFC	UHPFC	-
	UHPFC-RA	UHPFC	RA
	UHPFC-PA	UHPFC	PA
	UHPFC-OA	UHPFC	OA
	UHPFC-PT	UHPFC	PT
	UHPFC-OT	UHPFC	OT

**Table 2 materials-16-00072-t002:** Weight in kilograms of components for preparing a cubic meter of each type of concrete.

Type	Cement	Fly Ash	Silica Fume	Sand	Crushed Basalt	Fibers	Water	Superplasticizer
NC	332	-		662	830	-	206	-
HSC	460	80	54	600	980	-	130.7	9.2
SHCC	1300	-	230	146		15 PPF*	297	30.0
UHPFC	900	-	220	1005		157 SF*	162.4	40.3

PPF*: Polypropylene fiber, SF*: Steel fiber.

**Table 3 materials-16-00072-t003:** Summary of experimental results.

Sample	Δ_y_ (mm)	Δ_u_ (mm)	P_cr_*(kN)	P_y_* (kN)	P_u_ (kN)	Ductility (m)	P_u_/P_Control_	P_u_/(√ *f_c_’* × b0 × d)	Failure Mode
NC	-	0.25	13.40	-	75.10	-	1	0.332	Punching
NC-RA	-	0.30	14.10	-	84.60	-	1.13	0.374	Punching
NC-PA	-	0.30	14.60	-	88.65	-	1.18	0.392	Punching
NC-OA	-	0.31	15.60	-	96.90	-	1.29	0.428	Punching
NC-PT	-	0.35	14.40	-	96.25	-	1.28	0.426	Punching
NC-OT	-	0.35	14.80	-	102.90	-	1.37	0.455	Punching
HSC	-	0.20	20.40	-	93.50	-	1	0.328	Punching
HSC-RA	-	0.26	21.10	-	104.65	-	1.12	0.367	Punching
HSC-PA	-	0.32	21.60	-	111.50	-	1.19	0.391	Punching
HSC-OA	-	0.30	22.60	-	121.40	-	1.30	0.425	Punching
HSC-PT	-	0.45	21.00	-	124.40	-	1.33	0.436	Punching
HSC-OT	-	0.35	21.30	-	131.30	-	1.40	0.460	Punching
SHCC	2.20	3.65	13.60	51.10	211.90	Δ_u_/Δ_y_ = 1.66	1	0.640	Flexural
SHCC-RA	1.30	3.30	14.16	50.90	252.40	2.54	1.19	0.770	Flexural
SHCC-PA	2.14	2.76	14.60	60.70	248.00	1.29	1.17	0.757	Flexural
SHCC-OA	0.64	2.85	15.40	46.10	270.60	4.45	1.28	0.820	Flexural
SHCC-PT	0.36	2.94	14.30	32.40	246.50	8.17	1.16	0.750	Flexural
SHCC-OT	0.41	2.90	14.70	36.90	271.10	7.07	1.28	0.820	Flexural
UHPFC	1.75	4.02	19.00	60.50	254.40	2.3	1	0.630	Flexural
UHPFC-RA	1.02	3.75	19.60	67.60	317.06	3.68	1.25	0.780	Flexural
UHPFC-PA	0.44	4.00	20.00	50.50	299.45	9.1	1.18	0.740	Flexural
UHPFC-OA	0.51	3.90	20.80	58.20	342.70	7.65	1.35	0.840	Flexural
UHPFC-PT	0.36	3.00	19.50	45.10	284.30	8.33	1.12	0.700	Flexural
UHPFC-OT	0.35	3.94	19.70	45.90	328.10	11.26	1.29	0.810	Flexural

P_cr_*: load at first crack. P_y_*: load at the first yield of steel reinforcement meshes.

**Table 4 materials-16-00072-t004:** CDP parameters used for concretes, mechanical properties of steel bar meshes, and steel skeleton.

Concrete							
Type	Elastic Modulus E (N/mm^2^)	Poisson Ratio (ν)	Dilation Angle (ψ)	Eccentricity (e)	Shape Parameter (K_c_)	(*f_bo_/f_co_*) Maximum Compression Axial/Biaxial	Viscosity (μ)
NSC	31,600	0.2	30°	0.1	0.667	1.16	0
HSC	55,000	0.2	30°				
SHCC	29,100	0.17	35°				
UHPFC	46,000	0.15	36°				
**Steel**							
**Type**		**Elastic Modulus E (N/mm^2^)**		**Poisson Ratio ν**		**Yield Stress** **(N/mm^2^)**	
Longitudinal reinforcement		200,000		0.3		413	
Steel sections		190,707		0.3		240	
Bolts		200,000		0.3		900	

**Table 5 materials-16-00072-t005:** Difference between numerical model and experimental results.

Specimen	Max. FE Displacement Δ_u FE._ (mm)	Max. EXP. Displacement Δ_u EXP._ (mm)	Δ_u FE._/Δ_u EXP_.	Max. FE Failure Load P_u FE._ (kN)	Max. EXP. Failure Load P_u EXP._ (kN)	P_u FE._/P_u EXP._	V_u an._*	V_u an._/P_u EXP._
NC	0.27	0.25	1.08	75.90	75.10	1.01	71.20	0.95
NC-RA	0.32	0.30	1.07	85.70	84.60	1.01	82.30	0.97
NC-PA	0.33	0.30	1.10	89.25	88.65	1.01	85.30	0.96
NC-OA	0.33	0.31	1.06	98.25	96.90	1.01	92.15	0.95
NC-PT	0.36	0.35	1.03	96.80	96.25	1.01	94.16	0.98
NC-OT	0.37	0.35	1.06	105.40	102.90	1.02	101.3	0.98
HSC	0.21	0.20	1.05	96.50	93.50	1.03	89.50	0.96
HSC-RA	0.24	0.26	0.92	107.8	104.65	1.03	99.40	0.95
HSC-PA	0.40	0.32	1.25	113.70	111.50	1.02	103.20	0.93
HSC-OA	0.42	0.30	1.25	125.10	121.40	1.03	115.30	0.95
HSC-PT	0.51	0.45	1.13	131.00	124.40	1.05	117.20	0.94
HSC-OT	0.40	0.35	1.14	127.20	131.30	0.97	121.30	0.94
SHCC	3.20	3.65	0.88	215.20	211.90	1.02	195.90	0.92
SHCC-RA	3.45	3.30	1.05	259.10	252.40	1.03	230.20	0.91
SHCC-PA	3.17	2.76	1.15	256.40	248.00	1.03	230.40	0.93
SHCC-OA	3.20	2.85	1.12	279.70	270.60	1.03	260.15	0.96
SHCC-PT	2.95	2.94	1.01	256.50	246.50	1.04	235.15	0.95
SHCC-OT	2.83	2.90	0.98	276.20	271.10	1.02	262.40	0.97
UHPFC	3.70	4.02	0.92	260.80	254.40	1.03	245.20	0.96
UHPFC-RA	4.10	3.75	1.09	325.30	317.06	1.03	313.15	0.99
UHPFC-PA	3.70	4.00	0.93	303.90	299.45	1.02	285.20	0.95
UHPFC-OA	3.58	3.90	0.92	348.30	342.70	1.02	345.15	1.00
UHPFC-PT	3.50	3.00	1.17	290.80	284.30	1.02	275.20	0.97
UHPFC-OT	4.00	3.94	1.02	333.70	328.10	1.02	320.60	0.98

V_u an._*: analytical failure load.

## Data Availability

Not applicable.

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
