# Peer review of "Punching Shear Behavior of Slabs Made from Different Types of Concrete Internally Reinforced with SHCC-Filled Steel Tubes"

_materials, 2022, doi:10.3390/ma16010072_

Round 1
Reviewer 1 Report
This paper presents an experimental and numerical study to explore the behavior of flat slabs made of different types of concrete under the influence of punching shear and reinforced internally by high-strength steel bolts or SHCC-filled steel tubed sections. The topic is worth of investigation and the paper is of interests to the readers. The paper can be improved and the comments are listed below.
1. Design of the test specimens in one of the main issues in this manuscript. The applicability of the results from this system and the size effect contribution to larger-scale or real structural member is doubtful.
2. The material properties of fibers should be given. For SHCC and UHPFC, it is better to provide tensile stress-strain curves. The test setup and examples of tensile stress-strain curves can be seen in Eng. Struct. 272 (2022) 115020.
3. The use of shear reinforcement effectively depends on the end anchors (ie. studs, headed bars, closed stirrups) to provide confinement and to control crack propagation. The configuration of the shear reinforcement may not work as designed. Besides this system is challenging in construction. Please explain.
4. The test setup shows that the column is very long. What was the height of the column? This will lead to accentricity to the loading, and the eccentricity could alter the failure mode. Please discuss.
5. Comprehensive discussions should be provided on failure modes as well as key test results.
6. Please suggest a design approach for punching shear strength.
Author Response
Please check the attached file. Thank you for your good comments.

Reviewer 2 Report
The authors have studied the punching shear resistance of the slab by incorporating steel tubes filled with strain hardening cementitious concrete (SHCC)
I have following comments on the work
1. Shear strength is one of the most important property depends upon concrete as well as steel properties (Vc + Vs). I cannot understand the basic concept of the work, why we need to fill the steel tubes with SHCC? to improve the shear strength . Justification of this question is very important before publication of the work. Rest, increase or decrease is then meaningless
Means basic idea of the paper has a big question which need to be justified and answered prior to the acceptance
2. Figure 5 to 8 are not clear, please provide some clear images, and state what you want to show/present from these figures
3. Figure 9 shows typical results, please note if you conduct punching shear test on same class of concrete, this difference in the results will be still there
4. Table 3, Ductility. how you measure the ductility ? what is the purpose of presenting the ductility in case of shear resistance test?
5. Figure 16, which constitutive model you have used for the input properties of the materials?
6. Figure 18 totally not clear, (so much information is provided at the same time) what we need to get out of it? what is purpose ?
7. what kind of boundary conditions were used during the modeling?
8. how SHCC effect was considered during the modeling?
9. if we consider viscosity parameter 0.001 ( or similar) what will be effect on the results?
10. Line 202 (CDP) concrete damage plasticity model, where are the input concrete and steel constitutive relationships? (consider this comment with comment 5)
Please try to answer above questions and queries, my recommendation is Major Revision (Answer of Question is very important).
Author Response

(The authors gave the same response as above.)

Round 2
Reviewer 1 Report
The revised version can be accepted.
Author Response
Reviewer #1 The authors would like to thank the reviewers for the time and effort they devoted to reviewing our paper. An itemized list of the author's responses to the comments raised by the reviewers is provided below. The corrections or modifications are colored ( in red ) in the new version of the paper to ease revision. The revised version can be accepted. The reply: Thank you very muchReviewer 2 Report
Many Thanks for the resubmission of the Manuscript. I feel sorry to mention that I am not satisfied with the provided response. In my opinion There are some major technical issues in the paper needed to be addressed therefore I am not in the favor to recommend acceptance of the article at this stage. Some of the sample issues are provided here.
Refer to Line 69-71 shown below too
what do you mean by "to resist punching shear strength"?
Provide "Confinement" to what?
2. The explanation provided by the authors against my previous comments is also non satisfactory. Using steel tubes as internal reinforcement is still not clear, why authors have used steel filled tubes?
3. Line 181-182 ductility calculation in case of punching shear is still not clear. The response provided by the authors is not satisfactory
